# The Epidemiological, Clinical, and Microbiological Features of Patients with *Burkholderia pseudomallei* Bacteraemia—Implications for Clinical Management

**DOI:** 10.3390/tropicalmed8110481

**Published:** 2023-10-24

**Authors:** Carmen Prinsloo, Simon Smith, Matthew Law, Josh Hanson

**Affiliations:** 1College of Medicine and Dentistry, James Cook University, Cairns Hospital, Cairns, QLD 4870, Australia; 2Department of Medicine, Cairns Hospital, Cairns, QLD 4870, Australia; Simon.Smith2@health.qld.gov.au; 3The Kirby Institute, University of New South Wales, Sydney, NSW 2042, Australia; mlaw@kirby.unsw.edu.au; 4Global and Tropical Health Division, Menzies School of Health Research, Charles Darwin University, Darwin, NT 0811, Australia

**Keywords:** melioidosis, *Burkholderia pseudomallei*, bacteraemia, sepsis, critical care, clinical management, tropical medicine, antibiotic therapy, indigenous health, Australia

## Abstract

Patients with melioidosis are commonly bacteraemic. However, the epidemiological characteristics, the microbiological findings, and the clinical associations of *Burkholderia pseudomallei* bacteraemia are incompletely defined. All cases of culture-confirmed melioidosis at Cairns Hospital in tropical Australia between January 1998 and June 2023 were reviewed. The presence of bacteraemia was determined and correlated with patient characteristics and outcomes; 332/477 (70%) individuals in the cohort were bacteraemic. In multivariable analysis, immunosuppression (odds ratio (OR) (95% confidence interval (CI)): (2.76 (1.21–6.27), *p* = 0.02), a wet season presentation (2.27 (1.44–3.59), *p* < 0.0001) and male sex (1.69 (1.08–2.63), *p* = 0.02), increased the likelihood of bacteraemia. Patients with a skin or soft tissue infection (0.32 (0.19–0.57), *p* < 0.0001) or without predisposing factors for melioidosis (0.53 (0.30–0.93), *p* = 0.03) were less likely to be bacteraemic. Bacteraemia was associated with intensive care unit admission (OR (95%CI): 4.27 (2.35–7.76), *p* < 0.0001), and death (2.12 (1.04–4.33), *p* = 0.04). The median (interquartile range) time to blood culture positivity was 31 (26–39) hours. Patients with positive blood cultures within 24 h were more likely to die than patients whose blood culture flagged positive after this time (OR (95%CI): 11.05 (3.96–30.83), *p* < 0.0001). Bacteraemia portends a worse outcome in patients with melioidosis. Its presence or absence might be used to help predict outcomes in cases of melioidosis and to inform optimal clinical management.

## 1. Introduction

Melioidosis is a disease caused by the opportunistic, environmental Gram-negative bacterium *Burkholderia pseudomallei*. The organism is endemic in the tropics, with areas of hyperendemicity in northern Australia and Southeast Asia [1]. Melioidosis is strongly seasonal, is related to socioeconomic disadvantage and usually occurs in people with specific comorbidities that include diabetes mellitus, hazardous alcohol intake, chronic lung disease and chronic kidney disease [2,3,4,5].

The clinical presentation of melioidosis is highly variable: patients can present acutely in septic shock or after months of mild but persistent symptoms; the re-activation of latent disease, many years after infection, is also well-described [6]. Pneumonia is the most common clinical phenotype, but almost any organ can be involved. Organs that are classically involved include the skin and soft tissues, the bones and joints, the genito-urinary system—especially the prostate—the spleen, the liver, and the central nervous system (CNS). The majority of patients are bacteraemic, and the presence of bacteraemia portends a more complicated course [6,7,8]. However, despite the ubiquity of bacteraemia in patients with melioidosis, the epidemiological, clinical, and microbiological features of *B. pseudomallei* bacteraemia are rarely described in detail.

Australian guidelines for the antibiotic therapy of melioidosis, developed in the Northern Territory, use the different clinical phenotypes of melioidosis to inform the total duration of recommended treatment [9]. Antibiotic therapy is provided in two phases: the intensive phase—which aims to prevent death acutely—comprises of 2 to 8 weeks of intravenous meropenem or ceftazidime, and the eradication phase—which aims to prevent disease relapse—consists of 3 to 6 months of oral therapy, with trimethoprim-sulfamethoxazole (TMP-SMX) as the preferred agent [10]. These long courses of antibiotics have a salutary effect on survival in Australia and are associated with relapse rates that are lower than in other countries [8,9,11]. However, they are associated with a significant rate of drug-related side effects and there is increasing interest in high-volume centres in prescribing shorter courses of therapy [12,13].

The presence of bacteraemia is used to inform the duration of antibiotic therapy in a variety of infections [9,10,14,15,16]. However, in patients with melioidosis, a positive blood culture has a limited impact on the duration of therapy. Although a 2015 Australian study suggested that clinicians consider extending the intensive phase of antibiotics if blood cultures remain positive at seven days, there is no consensus on just how long this extension should be [17]. Nor are there any clear evidence-based recommendations on how the presence of bacteraemia might guide other aspects of clinical management, including approaches to imaging, source control and the use of adjunctive antimicrobial therapy.

This study aimed to examine the epidemiological, clinical, and microbiological characteristics of *B. pseudomallei* bacteraemia. It was hoped that the identification of factors associated with bacteraemia in cases of melioidosis might be used to inform the clinical management of patients with—and at risk of—this life-threatening disease.

## 2. Materials and Methods

This retrospective study was performed at Cairns Hospital, a 531-bed tertiary referral hospital in Far North Queensland (FNQ), tropical Australia. The hospital serves a population of approximately 290,000 people, 17% of whom identify as Aboriginal or Torres Strait Islander Australians, hereafter respectfully referred to as First Nations Australians.

Participants were eligible for inclusion in the study if they were managed in the local health service and had a positive culture for *B. pseudomallei* in the Cairns Hospital laboratory between the 1st of January 1998 and the 1st of June 2023. This period was chosen as it coincided with the establishment of a state-wide electronic laboratory database. The isolates of *B. pseudomallei* were identified by experienced scientists working in the Cairns Hospital laboratory, who cultured the organism using traditional and selective media. The results of antimicrobial susceptibility testing usually suggested *B. pseudomallei,* although identification was confirmed by MicroScan (Beckman Coulter, Brea, CA, USA) and API 20NE (bioMérieux, Marcy-l’Étoile, France) before 2004 and by the Vitek 2 system (bioMérieux, Marcy-l’Étoile, France) after this date.

Data were collected prospectively from October 2016 and retrospectively prior to this date. Electronic and paper hospital records were reviewed and demographic, clinical, laboratory and outcome data were recorded. Specific risk factors for melioidosis—diabetes mellitus, hazardous alcohol use, chronic lung disease, chronic kidney disease, immunosuppression, and active malignancy—were sought and recorded as described previously [8]. If individuals lacked any of these six comorbidities, they were said to have no risk factors for melioidosis. Children were defined as those less than 16 years of age. First Nations Australian status was collected from hospital records as every individual admitted to a public hospital in Queensland is asked if they identify as either Aboriginal, Torres Strait Islander, both or neither. If patients lived in the neighbouring Torres and Cape Hospital and Health Service—a region that comprises the Cape York Peninsula and the Torres Strait Islands—they were said to have a remote residence. Clinical phenotypes of melioidosis were also defined based on imaging findings and culture of *B. pseudomallei* from relevant specimens. Death attributable to melioidosis was recorded, as were disease recrudescence and relapse as defined previously [8].

The time to positivity (TTP) for a blood culture was the duration of time that a blood culture specimen spent incubating in the hospital’s automated blood culture system (BacT/ALERT^®^ bioMérieux, Marcy-l’Étoile, France) before bacterial growth was recorded. TTP was only recorded for blood cultures taken within Cairns Hospital, as blood cultures from satellite hospitals were frequently not incubated until several hours after collection. Duration of bacteraemia was defined as the time between the collection of the first positive blood culture and the collection of the last positive blood culture (if a patient had multiple positive blood cultures). Source control was defined as any drainage performed by a surgeon in an operating theatre or an interventional radiologist. Population data from the Australian Bureau of Statistics were used to determine disease incidence.

Data were de-identified, entered into an electronic database (Microsoft Excel 2016, Microsoft, Redmond, WA, USA) and analysed using statistical software (Stata version 14.2, StataCorp LLC., College Station, TX, USA). Univariate analysis was performed using Chi-squared tests and logistic or linear regression, as appropriate. Multivariable analyses used logistic regression for binary endpoints, linear regression for continuous endpoints, and a backwards stepwise approach with variables selected for consideration in the multivariable model if their *p*-value in univariate analyses was <0.10. To avoid omitting observations with missing data in the multivariable models—which would have led to excluding up to 30% of the data—we created missing data categories. Trends over time were determined using logistic regression or an extension of the Wilcoxon rank sum test, where appropriate [18]. To retain power, no adjustments were made to significance levels for multiple comparisons. A correlation matrix was constructed to examine the relationship between explanatory variables (Appendix A).

The study was conducted in accordance with the Declaration of Helsinki, and the protocol was approved by The Far North Queensland Human Research Ethics Committee, which provided ethical approval for the study (HREC/18/QCH/91–1261 and HREC/15/QCH/46–977). As the data were de-identified, the Committee waived the requirement for informed consent.

## 3. Results

Over the 25-and-a-half-year study period, 477 patients were diagnosed with culture-confirmed melioidosis. The incidence of melioidosis in the FNQ region increased from 4.6/100,000 in 1998 to 20.4/100,000 in 2022—the first and last complete calendar year in the study period (*p* for trend = 0.001).

The patients’ median (interquartile range (IQR)) age was 54 (42–65) years, 333/477 (70%) were male, and 231/477 (48%) identified as First Nations Australians. The most common risk factors for melioidosis in the cohort were diabetes and hazardous alcohol intake; lung involvement was the most frequent clinical phenotype (Table 1). Overall, 55/477 (12%) died from their *B. pseudomallei* infection; the case-fatality rate (CFR) decreased from 13/45 (29%) in the first 5 years to 21/235 (9%) in the last 5 years of the study period (*p* for trend = 0.002). There were 10 cases of recrudescence, and 1 of these patients died. There were 10 cases of disease relapse, and 1 of these patients died.

### 3.1. Bacteraemia

Bacteraemia was confirmed in 332/477 (70%) patients (Table 1). There was no change in the incidence of bacteraemia over the course of the study (*p* for trend = 0.50). In the first 5 years of the study, 31/45 (69%) of the patients were bacteraemic compared to 154/235 (66%) in the last 5 years of the study. Associations with bacteraemia in univariate analysis are presented in Table 1. In multivariable analysis, five factors were independently associated with the presence of bacteraemia: immunosuppression (odds ratio (OR) (95% confidence interval (CI)): 2.76 (1.21–6.27), *p* = 0.02), a wet season presentation (OR (95% CI): 2.27 (1.44–3.59), *p* < 0.0001) and male sex (OR (95% CI): 1.69 (1.08–2.63), *p* = 0.02) increased the likelihood of bacteraemia. Patients with a skin or soft tissue infection (OR (95% CI): 0.32 (0.19–0.57), *p* < 0.0001) and those without predisposing factors for melioidosis (OR (95% CI): 0.53 (0.30–0.93), *p* = 0.03) were less likely to be bacteraemic.

Bacteraemia was associated with the presence of septic shock (OR (95% CI): 6.49 (3.05–13.82), *p* < 0.0001), intensive care unit (ICU) admission (OR (95% CI): 4.27 (2.35–7.76), *p* < 0.0001), and death due to melioidosis (OR (95% CI): 2.12 (1.04–4.33), *p* = 0.04).

### 3.2. Time to Positivity

There were 193 bacteraemic patients who had their initial blood cultures collected at Cairns Hospital. The median (IQR) TTP was 31 (26–39) hours (Figure 1). There was no change in the TTP over the course of the study period (*p* for trend = 0.43). In the first 5 years of the study period, the median (IQR) TTP was 33 (24–73) hours, compared with 30 (20–37) hours in the last 5 years of the study. There were 21/193 (11%) patients with positive blood cultures within 24 h, while 171/193 (89%) and 189/193 (98%) had positive blood cultures within 48 h and 72 h, respectively (Table 2). Patients with positive blood cultures within 24 h were 11 times more likely to die than patients whose blood cultures flagged positive later (OR (95% CI): 11.05 (3.96–30.83), *p* < 0.0001).

Overall, patients who died had a shorter median (IQR) TTP than the patients who survived (25 (22–31) versus 31 (27–40) hours, *p* = 0.0007). Patients with septic shock had a shorter median (IQR) TTP than those without septic shock (26 (23–33) versus 33 (28–40) hours, *p* = 0.0001). Patients admitted to the ICU had a shorter median (IQR) TTP than those not requiring ICU admission (28 (23–33) vs. 33 (28–40) hours, *p* = 0.0002). There were 54 patients requiring ICU admission who had blood cultures collected in Cairns and 48 (89%) were bacteraemic; 45 (94%) had their blood cultures flag positive within 48 h.

Lung involvement and wet season presentation were associated with a shorter TTP (Table 3). This was at least partly explained by the association between lung involvement and a wet season presentation (261/347 (75%) of patients presenting in the wet season had lung involvement, compared with 75/119 (63%) of patients presenting in the dry season, *p* = 0.01). There were 48 patients with lung involvement requiring ICU admission who had blood cultures collected in Cairns and 42 (88%) were bacteraemic; 38 (90%) had their blood cultures flag positive within 48 h.

In contrast, patients with genito-urinary involvement and those with musculoskeletal involvement had a longer TTP. In multivariable analysis, only musculoskeletal involvement (mean difference (95% CI): 3.9 (1.1 to 6.7) hours, *p* = 0.007) and genito-urinary involvement (mean difference (95% CI): 3.2 (0.6 to 5.8) hours, *p* = 0.02) were independently associated with TTP.

### 3.3. Duration of Bacteraemia

Of the 332 bacteraemic patients, 268 (81%) had blood cultures collected across multiple days. Of the 332 bacteraemic patients, 184 (55%) had multiple positive blood cultures. The median (IQR) duration of bacteraemia in these 184 patients was 42 (17–97) hours and it was 56 (18–98) hours in the 159 patients who survived their infection (Figure 2). There was no change in the duration of bacteraemia during the study period (*p* for trend = 0.50). The median (IQR) duration of bacteraemia among patients with recrudescence or relapse (71 (52–103) hours) was greater than that in those who did not develop these complications (40 (17–97) hours) but this difference did not reach statistical significance (*p* = 0.19). In multivariable analysis, male sex (mean difference (95% CI): 47 (9–86) hours, *p* = 0.02) and musculoskeletal involvement (mean difference (95% CI): 43 (4–83) hours, *p* = 0.03) were independently associated with a longer duration of bacteraemia (Table 4).

### 3.4. Source Control

Of the 477 patients in the cohort, 140 (29%) had source control but there was no increase in source control procedures over the course of the study (*p* for trend = 0.75). Among the 332 bacteraemic patients 89 (27%) had source control, compared to 51/145 (35%) of the non-bacteraemic patients (*p* = 0.07) (Table 5). Bacteraemia was present in 45/55 (82%) of the deaths, with death often occurring early in the patients’ hospitalisation (at a median of 4 (0–13) days after admission) which may have precluded source control in many. However, in multivariable analysis, source control was less common in patients with bacteraemia (OR (95% CI): 0.54 (0.33–0.88), *p* = 0.01), advancing age (OR (95% CI): 0.90 (0.85–0.97), *p* = 0.003) and an absence of risk factors for melioidosis (OR (95% CI): 0.41 (0.20–0.83), *p* = 0.01). Conversely, source control was more common in patients with musculoskeletal involvement (OR (95% CI): 8.41 (4.43–15.97), *p* < 0.0001) and genito-urinary involvement (OR (95% CI): 5.89 (3.47–9.98, *p* < 0.0001).

### 3.5. Impact of Duration of Bacteraemia on Source Control

Among the 143 surviving patients with multiple positive blood cultures, 47 (33%) had source control. The median (IQR) duration of bacteraemia in patients with source control was longer than that in patients who did not have source control (73 (25–146) hours versus 36 (15–73), *p* = 0.001), suggesting that persisting bacteraemia prompted clinicians to perform the source control.

## 4. Discussion

Bacteraemia is common in patients with melioidosis, and it portends a worse outcome. Almost 70% of individuals with melioidosis in this series had confirmed bacteraemia, and these patients were more than twice as likely to die from the infection than their non-bacteraemic counterparts. However, despite the ubiquity and, frequently, the persistence of bacteraemia in patients with melioidosis, the implications for clinical management are incompletely defined. It is unclear, in particular, what impact the presence and duration of bacteraemia might have in informing patients’ diagnostic work-up, adjunctive care and more tailored, individualised approaches to the duration of their antibiotic therapy [19,20,21].

Although the presence of bacteraemia was associated with septic shock, ICU admission, and a significantly increased risk of death, these high-risk patients are likely to be identified by the derangement of other more easily collected clinical and laboratory indices, which are more accessible in the rural and remote and low- and middle-income settings that have a greater burden of the disease [22,23,24,25]. Bacteraemia clearly has prognostic utility for a variety of pathogens [26]; however, it is notable that the median TTP of *B. pseudomallei*—31 h in this series—is significantly longer than for other common Gram-negative pathogens [27].

In this context, it is significant that patients with positive blood cultures in the first 24 h of their hospitalisation were more than 11 times as likely to die than bacteraemic patients whose blood cultures flagged later in their admission. Moreover, even in Australia’s well-resourced health system, death occurred in almost half of the patients with positive blood cultures in the first 24 h of their hospitalisation. The prognostic value of a shorter TTP was also demonstrated in a 1997 Thai study that reported a CFR of 74% in patients with a TTP of <24 h—almost twice the CFR of bacteraemic patients with a TTP of >24 h [28]. Another Australian study identified that TTP was associated with septic shock and ICU admission, although in this smaller study, the association with death did not reach statistical significance [29]. A shorter TTP has been shown to have prognostic utility in other life-threatening infections and may represent a higher bacterial load in the patient at the time blood cultures were taken [30,31,32]. In this context, it was notable that the TTP was generally shorter in patients with pulmonary involvement, where multi-lobar involvement and abscesses are common [33]. The lungs are also highly vascular, which may be one explanation for the fact that, conversely, patients with involvement of less vascular sites (musculoskeletal involvement—which was often osteomyelitis—and genito-urinary involvement—which was often prostatic involvement) had a longer TTP.

Persistent bacteraemia in patients with melioidosis may suggest an unrecognised focus of infection—classically hepatic, splenic or prostatic abscesses or osteomyelitis—that may require dedicated, sophisticated imaging (such as CT, magnetic resonance imaging and positron emission tomography) for detection [34]. Rarely it can be evidence of a mycotic aneurysm, a feared complication of melioidosis with a high attributable mortality rate, which usually requires surgical intervention and longer—sometimes lifelong—courses of antibiotic therapy [35,36]. It was notable that almost 30% of patients in this series were able to have source control, and this may have contributed to the cohort’s favourable outcomes and low rates of recrudescence and relapse [11]. Source control was actually less common in bacteraemic patients in this cohort, emphasising the importance of source control in the optimal management of patients with infection [37,38], although this may have been partly explained by the fact that older patients—with greater comorbidity—were not well enough to undergo source control procedures and that sicker patients did not survive long enough to have these procedures [39]. Indeed, persistent bacteraemia was often a prompt for source control in our cohort, which was performed in more than half the patients with musculoskeletal involvement (which commonly represented debridement of osteomyelitis) and genito-urinary involvement (which commonly represented drainage of prostate abscesses). Early source control is at least partly responsible for the excellent outcomes seen in Australian patients with these manifestations [40,41]. Persistent bacteraemia might also prompt extended antimicrobial resistance testing and the addition of adjunctive antibiotic therapy [10].

While the CFR of melioidosis in low- and middle-income countries is still greater than 35% [7,42,43,44,45], it has decreased dramatically in Australia over the last 30 years and is now less than 10% in high-volume centres [6,46]. This is likely due to information campaigns that have educated the population and clinicians about the early signs of sepsis; access to sophisticated supportive care in Australia’s well-resourced health system, and the development and promulgation of evidence-based guidelines for the antibiotic therapy of individuals at risk for melioidosis [10,47,48,49].

The duration of this antibiotic therapy is informed, almost entirely, by the site of the infection [10]. However, clinicians in high-volume centres are interested in developing a more individualised duration of therapy, which could be guided not only by the site of infection but also the presence and duration of bacteraemia, the ability to provide source control and the reversibility of risk factors for the disease (such as poor glycaemic control in patients with diabetes or hazardous alcohol consumption) [13]. It may be that the routine, regular collection of blood cultures after the diagnosis of melioidosis until they are negative—as occurs in cases of *Staphylococcus aureus* bacteraemia—could help inform the optimal duration of antibiotic therapy while also prompting ancillary management strategies and assisting with prognostication [21,36]. Indeed, this was a strategy proposed by Thai investigators who demonstrated that persistent bacteraemia was associated with an increase in the risk of death in a 2011 study [45].

The absence of *B. pseudomallei* bacteraemia in patients presenting for care may also assist clinical management in areas that are endemic for melioidosis. Australian guidelines recommend that selected patients presenting with septic shock or severe pneumonia in tropical Australia should have meropenem included in their empirical antibiotic regimen. This is primarily to cover the possibility of melioidosis, as it is a common cause of sepsis—particularly respiratory sepsis—in the region [50,51]. However, the fact that almost 90% of the blood cultures flagged positive within 48 h—and that almost 75% of patients with pneumonia were bacteraemic—may be used to inform decisions about de-escalation in selected patients. This de-escalation is imperative for countries in Southeast Asia, where melioidosis is also endemic, and where carbapenem-resistant Enterobacterales (CRE) are common and the responsible use of carbapenems is essential [14,52].

Bacteraemia was detected in 70% of patients in this FNQ cohort, the same rate of bacteraemia that was reported in a 2021 retrospective study from Townsville Hospital in North Queensland, which is about 350 km south of Cairns [53]. This figure is significantly greater than the rate of 56% reported in the landmark 30-year Darwin Prospective Melioidosis Study (DPMS) [6]. While the rates of predisposing comorbidities are generally similar between these high-volume Australian centres, there are some differences [46]. It was notable that immunosuppression—which was one of only three factors to be independently associated with an increased rate of bacteraemia—was twice as common in our cohort than in the DPMS (18% vs. 9%) [6]. There may also be differences in virulence factors between the regions—a possibility that has been hypothesised to explain the differences in the clinical phenotype among children with melioidosis in the two locations [54,55,56].

Most patients diagnosed with melioidosis outside of Australia are also bacteraemic. Depending on the study, the rates of bacteraemia range from approximately 58% to 77% in Thailand [7,45,57]; from about 55% to 61% [58,59] in India; from about 77% to 88% [43,60] in Malaysia; and from 60% to 68% [61,62] in Singapore; and they are as high as 73% in Sri Lanka [63]. An explanation for the apparent difference in the bacteraemia rates might be explained by differences in clinical culturing practices between the sites, differences in rates of comorbidities, or differences in virulence factors of local strains [56]. The variation between these countries may also be explained by differences in patients’ access to healthcare at different sites. Socioeconomic, cultural, and geographical factors will impact the timing of patients’ presentation, the prevalence—and management—of predisposing medical comorbidities, and the resources for laboratory and radiological investigations that might facilitate a diagnosis of melioidosis. There may be variations in the proportion of different populations who have blood cultures collected to identify melioidosis. Patients admitted to the ICU in well-resourced health systems are likely to have multiple blood cultures collected [49,64]. In contrast, blood cultures were only collected in 44% of the children in a 2021 Cambodian study to examine the disease [65]. Self-limiting skin lesions might be managed entirely in the community, almost always without blood cultures and potentially without antibiotics, and therefore would not be included in hospital-based studies.

Why some individuals with melioidosis develop bacteraemia and others do not is incompletely understood but is likely to be explained by the size of the inoculation, the site of the inoculation, and differences in host susceptibility. There may also be a contribution of virulence factors that are present in different *B. pseudomallei* strains [4]. Melioidosis is uncommon in the absence of well-described risk factors, which is likely to explain the lower rate of disease in children [55,66]. However, the disease is well documented in otherwise well individuals after near drowning or exposure to contaminated medical and commercial products [67,68,69]. The size and site of the inoculum appear to play a role in the development of the disease and its complications. A murine model of melioidosis infection demonstrated that the lethal dose of *B. pseudomallei* was much higher via an enteral (7 × 10^8^ colony forming units (CFU)) as compared to an inhaled route (5 × 10^2^ CFU) [70]. Almost 75% of patients with lung involvement in this cohort were bacteraemic, and while the respiratory tract is a common source of bacteraemia with other pathogens, the rate of bacteraemia is usually less than 10%, even among inpatients [71,72]. Why the rate of bacteraemia is higher in patients with melioidosis pneumonia is incompletely understood, although the volume of infected tissue and the behaviour of *B. pseudomallei* in the milieu of the lung may be important [73]. The association in this cohort between wet season presentation and lung involvement in melioidosis is well recognised and may reflect a shift towards inhalation as the mode of acquiring *B. pseudomallei* [5,74]. Patients presenting after heavy rainfall in the monsoonal wet season in the Northern Territory of Australia were twice as likely to have bacteraemic pneumonia, which has been hypothesised to represent the inhalation of a larger inoculation dose [74]. This association between a wet season presentation, pneumonia and bacteraemia was seen again in our cohort.

The innate and adaptive immune response both play an important role in resistance to *B. pseudomallei* infection [1,3], and immunosuppression had the strongest independent association with bacteraemia in this cohort and the DPMS [6]. In contrast, an absence of risk factors for melioidosis, which are believed to impair immunity to *B. pseudomallei* [4], also reduced the likelihood of bacteraemia in our cohort. Although diabetes mellitus was the most common predisposing factor for melioidosis in our series, and the most common risk factor for melioidosis globally, it was not associated with bacteraemia as it was in other large cohorts from Australia and Thailand [6,75]. This might be explained by the fact that bacteraemia was more common in our cohort than in these other series. In contrast, bacteraemia was much less common in patients with SSTI in both our and other cohorts, reflecting the elements of the innate and adaptive immune system that are present in the skin that prevent inoculated bacteria from disseminating into the bloodstream [6,53]. The explanation for the independent association between male sex and bacteraemia is unclear but may relate to differences in health-seeking behaviour—data that were not captured in this analysis [76,77]. Men also had higher rates of hazardous alcohol use and tobacco smoking that were both associated with pneumonia in univariate analysis, although a more detailed comparison of gender differences in lifestyle and the social determinants of health was not possible in this—at least partly—retrospective study.

There are several virulence genes that have been linked to the clinical phenotype and clinical course of patients with melioidosis [4]. The presence of the *Burkholderia mallei*-like *BimA_Bm_* virulence gene, for instance, is associated with CNS melioidosis presentations and death and long-term disability [78]. The development of bacteraemia has been linked most strongly to the presence of a filamentous hemagglutinin gene, *fhaB3*, which encodes for FhaB3, an anti-macrophage factor involved in host epithelial cell attachment. A 2014 study from the Northern Territory of Australia found that 83% of isolates had the *fhaB3* gene, and patients infected with these isolates were twice as likely to be blood-culture-positive. In contrast, patients with *fhaB3*-negative isolates were four times more likely to have an SSTI without sepsis [56]. Although the isolates in our study were not tested for the presence of the *fhaB3* gene, it is possible that the presence—or absence—of this gene may have contributed to the inverse association between SSTI and bacteraemia that was seen in the cohort.

This study has several limitations. The study’s retrospective nature precluded comprehensive data collection prior to 2016. The absence of a standardised approach to blood culture collection after the diagnosis of melioidosis also limits the reliability of the analysis of the duration of bacteraemia. It is clinical practice to take blood cultures in patients who are deteriorating clinically, which is likely to mean that patients with a more complicated course were over-represented in our analysis of the duration of bacteraemia. The study spanned a period of over 25 years, which coincided with an evolution in guidelines for the management of melioidosis, advancements in critical care, and better access to diagnostic imaging, which all impacted the clinical course of the patients. A tenfold increase in the local incidence of melioidosis during the study period also resulted in greater clinical awareness of the infection and therefore earlier recognition, which almost certainly impacted the clinical course of the patients [79]. Changes in the clinical management of sepsis over the course of the study period—particularly an emphasis on the collection of blood cultures and recommendations for early antibiotic therapy [38]—may, in theory, have impacted the rate of bacteraemia, the TTP and the number of patients who had data available on the duration of bacteraemia; however, there was no change in these variables over the course of the study. In order to retain power, we did not adjust significance levels for multiple comparisons, which increases the risk of family-wise error. However, this approach allowed for the examination of multiple potential associations in this exploratory analysis, which were then evaluated further in multivariable analysis. The provision of the raw data allows the reader—and other clinicians—to assess the biological plausibility of our findings. The study was performed in Australia’s well-resourced health system, limiting the generalisability of the findings and applicability of suggested management strategies to resource-limited settings with less access to sophisticated imaging and laboratory support. However, the prognostic utility of bacteraemia and the essential nature of prompt source control are almost certainly likely to be equally relevant.

Future studies might determine the utility of systematically collecting blood cultures in patients with melioidosis. These prospective studies could describe the association between the presence and duration of bacteraemia and the TTP to confirm the association with comorbidity, clinical phenotype and clinical course that was identified in this retrospective study. A larger study may be powered to identify the utility of these microbiological indices with less common clinical presentations such as CNS disease or endovascular infection, providing greater insights into the pathophysiology of melioidosis. Ultimately, however, it would be important to examine if the presence and duration of bacteraemia and TTP might be used to actually influence patient management. Specifically, whether these data might be used to inform adjunctive therapy, source control and duration of therapy, particularly short-course therapy. The precise factors that might allow treatment to be abbreviated—and the degree to which treatment can be shortened—remain to be determined [45,56,80]. However, as in other pathogens, the aim would be to reduce the morbidity and mortality associated with infection and the costs associated with unnecessarily prolonged therapy [21,81]. A systematic evaluation of the presence of different virulence factors and a multivariable analysis of their impact on patients’ clinical presentation and course would also be informative.

## 5. Conclusions

Most patients hospitalised with melioidosis will be bacteraemic (and bacteraemia portends worse outcomes), but the actual impact of bacteraemia on clinical management strategies remains incompletely defined. The regular collection of blood cultures, until they are negative, may inform optimal care, with persistent bacteraemia prompting the clinician to seek and address an unrecognised focus of infection. The duration of bacteraemia may also have utility in defining the duration of optimal antibiotic therapy, as occurs in other infections [82]. However, the clinical algorithms that might inform the optimal duration of antibiotic therapy remain to be determined in prospective studies. Importantly, although a comprehensive, directed approach to the acute presentation of melioidosis is essential to improve in-hospital survival, clinicians also need to consider the subsequent management of the patient and the underlying drivers of infectious diseases if we are to also reduce the long-term morbidity and mortality that results from this life-threatening pathogen [2,38,57,83].

## Figures and Tables

**Figure 1 tropicalmed-08-00481-f001:**
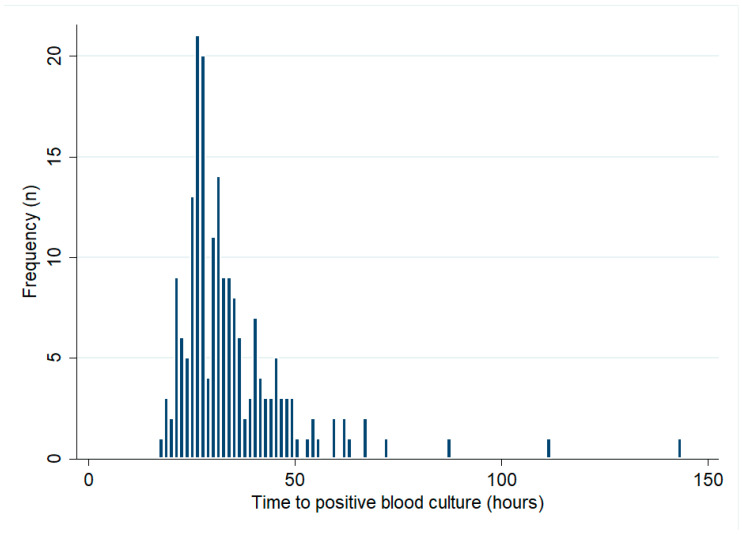
Blood cultures’ time to positive (only patients with blood cultures collected at Cairns Hospital are shown).

**Figure 2 tropicalmed-08-00481-f002:**
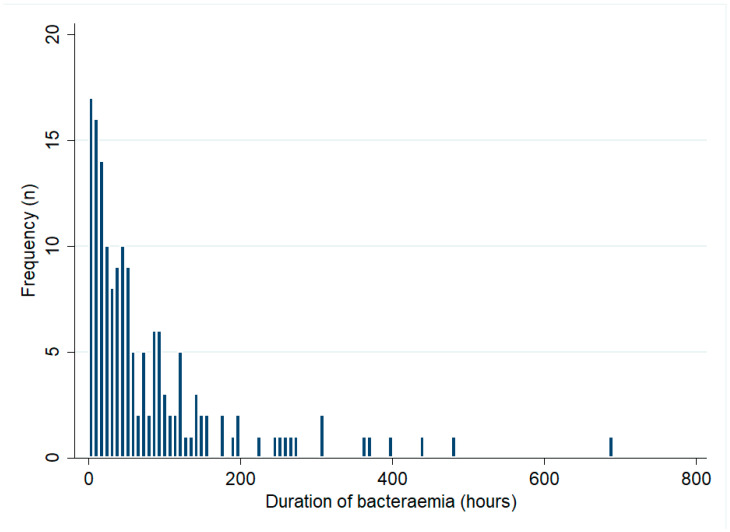
Duration of bacteraemia among patients with multiple positive blood cultures (only patients who survived to hospital discharge are shown).

**Table 1 tropicalmed-08-00481-t001:** Comparison of demographic and clinical characteristics of patients with melioidosis who did- and did not have bacteraemia.

	All *n* = 477 ^a^	Non-Bacteraemic *n* = 144	Bacteraemic *n* = 333	Odds Ratio (95% Confidence Interval)	*p* ^b^
Age (years)	54 (42–65)	50 (35–66)	56 (44–65)	1.08 (1.03–1.14) ^c^	0.004 ^d^
Child (<16 years)	23 (5%)	15 (10%)	8 (2%)	0.21 (0.09–0.52)	0.001 ^d^
Male sex	333 (70%)	88 (61%)	245 (74%)	1.74 (1.15–2.64)	0.008 ^d^
First Nations Australians	231 (48%)	64 (44%)	167 (50%)	1.23 (0.83–1.82)	0.30
Remote residence ^b^	171 (36%)	50 (35%)	121 (36%)	1.04 (0.69–1.57)	0.84
Wet season presentation	164 (74%)	90 (63%)	265 (80%)	2.42 (1.57–3.71)	<0.0001 ^d^
Diabetes mellitus	237/460 (51%)	67/138 (49%)	170/322 (53%)	1.16 (0.78–1.73)	0.46
Hazardous alcohol use	171/444 (39%)	40/131 (31%)	131/313 (42%)	1.59 (1.03–2.44)	0.04 ^d^
Current tobacco smoker	221/441 (50%)	58/132 (44%)	163/309 (53%)	1.42 (0.95–2.15)	0.09 ^d^
Chronic lung disease	95/447 (21%)	29/132 (22%)	66/315 (21%)	0.95 (0.58–1.56)	0.85
Chronic kidney disease	59/458 (13%)	12/138 (9%)	47/320 (15%)	1.83 (0.94–3.57)	0.08
Immunosuppression	58/327 (18%)	8/101 (8%)	50/226 (22%)	3.30 (1.50–7.26)	0.003 ^d^
Malignancy	47/446 (11%)	9/132 (7%)	38/314 (12%)	1.90 (0.89–4.06)	0.10 ^d^
No risk factors for melioidosis	73 (15%)	36 (25%)	37 (11%)	0.38 (0.23–0.63)	<0.0001 ^d^
Lung involvement	336/466 (72%)	85/137 (62%)	251/329 (76%)	1.94 (1.26–2.97)	0.002 ^d^
Genito-urinary involvement	90/460 (20%)	30/135 (22%)	60/325 (18%)	0.80 (0.49–1.31)	0.38
Musculoskeletal involvement	58/459 (13%)	12/135 (9%)	46/324 (14%)	1.72 (0.88–3.35)	0.11
SSTI	70/459 (15%)	40/135 (30%)	30/324 (9%)	0.25 (0.14–0.42)	<0.0001 ^d^
CNS involvement	16/459 (3%)	5/135 (4%)	11/324 (3%)	0.92 (0.31–2.71)	0.89
Septic shock	99/443 (22%)	8/132 (6%)	91/311 (29%)	6.49 (3.05–13.82)	<0.0001
ICU admission	118 (25%)	14 (10%)	104 (31%)	4.27 (2.35–7.76)	<0.0001
Died	55 (12%)	10 (7%)	45 (14%)	2.12 (1.04–4.33)	0.04
Relapse/recrudescence	20 (4%)	8 (6%)	12 (4%)	0.64 (0.26–1.61)	0.34

All numbers are presented as median (interquartile range) or the absolute number (%). SSTI: skin and soft tissue infection; CNS: central nervous system; ICU: intensive care unit. ^a^ Retrospective data collection from cases before October 2016 resulted in some missing data prior to this time and, accordingly, a difference in the denominator for some variables. ^b^ *p* value determined using univariable logistic regression. ^c^ Age (in years) divided by 5 for this analysis to facilitate interpretation. ^d^ Variables selected for the multivariable analysis.

**Table 2 tropicalmed-08-00481-t002:** Association between time to blood culture positivity and survival.

	Number	Survived	Died
TTP < 24 h	21	11 (52%)	10 (48%)
TTP ≥ 24 h and <48 h	150	138 (92%)	12 (8%)
TTP ≥ 48 h and <72 h	18	17 (94%)	1 (6%)
TTP ≥ 72 h	4	4 (100%)	0
Overall	193	170 (88%)	23 (12%)

TTP: time to positive: the time (in hours) that a blood culture specimen spent in the hospital’s automated blood culture system (BacT/ALERT^®^ bioMérieux) before bacterial growth was recorded.

**Table 3 tropicalmed-08-00481-t003:** Association between demographic clinical factors and the time to positive blood culture.

	*n* = 193 ^a^	TTP If Present (IQR) Hours	TTP If Not Present Median (IQR) Hours	TTP If Present (SD) Hours	TTP If Not Present Mean (SD) Hours	Mean Difference 95% Confidence Interval) Hours	*p* ^b^
Male gender	132	30 (26–39)	31 (26–40)	32 (6.4)	32 (6.0)	−0.3 (−2.6 to 1.9)	0.76
First Nations Australian	78	33 (26–41)	30 (26–37)	33 (6.2)	32 (6.2)	1.4 (−0.6 to 3.5)	0.18
Wet season presentation	147	30 (26–36)	37 (28–42)	32 (6.0)	34 (6.5)	−2.9 (−5.2 to −0.5)	0.02 ^c^
Diabetes mellitus	92	32 (26–42)	30 (27–37)	32 (6.6)	32 (6.0)	0.1 (−1.9 to 2.1)	0.92
Hazardous alcohol use	71	31 (26–40)	31 (26–39)	33 (6.7)	32 (6.1)	1.2 (−0.9 to 3.3)	0.26
Chronic kidney disease	31	33 (29–43)	30 (26–38)	34 (6.9)	32 (6.1)	1.8 (−0.7 to 4.4)	0.16
Chronic lung disease	49	31 (27–42)	31 (26–38)	32 (7.0)	32 (6.0)	0.2 (−2.1 to 2.4)	0.90
Immunosuppression	37	32 (25–40)	31 (27–39)	32 (6.4)	33 (6.3)	−0.3 (−3.0 to 2.4)	0.81
Malignancy	30	31 (26–37)	31 (27–40)	31 (5.4)	32 (6.4)	−1.3 (−4.0 to 1.5)	0.36
No risk factors	15	31 (28–34)	31 (26–40)	30 (2.9)	32 (6.5)	−2.5 (−6.0 to 1.1)	0.18
Pneumonia	148	30 (26–36)	36 (28–43)	32 (6.1)	34 (6.5)	−2.1 (−4.4 to 0.3)	0.08 ^c^
SSTI	16	34 (29–43)	31 (26–39)	34 (6.0)	32 (6.3)	1.7 (−2.1 to 5.4)	0.38
Musculoskeletal	23	36 (28–42)	30 (26–38)	35 (6.5)	32 (6.1)	3.6 (0.7 to 6.5)	0.01 ^c^
CNS involvement	6	34 (25–46)	31 (26–39)	35 (10.5)	32 (6.1)	2.7 (−2.4 to 7.9)	0.30
Genito-urinary involvement	34	33 (28–44)	31 (26–38)	35 (7.0)	32 (6.0)	2.9 (0.3 to 5.6)	0.03 ^c^
Isolated bacteraemia	23	37 (28–44)	31 (26–39)	33 (5.8)	32 (6.3)	1.0 (−2.9 to 4.9)	0.60

TTP: time to positive; IQR: interquartile range; SD: standard deviation. SSTI: skin and soft tissue infection; CNS: central nervous system. Median and mean are presented as the data had a non-parametric distribution. ^a^ Only includes the 193 bacteraemic individuals who had their initial blood culture collected in Cairns Hospital. ^b^ *p* value determined using linear regression. ^c^ Variables selected for the multivariable analysis.

**Table 4 tropicalmed-08-00481-t004:** Impact of demographics, comorbidities, and clinical phenotypes on the duration of bacteraemia among the 159 survivors with multiple positive blood cultures.

	*n* = 159 (%) ^a^	Median (IQR) Duration of Bacteraemia If Factor Present (Hours)	Median (IQR) Duration of Bacteraemia If Factor Not Present (Hours)	Mean (SD) Duration of Bacteraemia If Factor Present (Hours)	Mean (SD) Duration of Bacteraemia If Factor Not Present (Hours)	Mean Difference (95% Confidence Interval) (Hours)	*p* ^b^
Male gender	126 (79%)	50 (19–117)	39 (12–52)	90 (111)	43 (37)	48 (9–87)	0.02 ^c^
First Nations Australians	85 (53%)	46 (20–118)	45 (13–91)	94 (123)	65 (69)	29 (−3 to 61)	0.07 ^c^
Diabetes mellitus	92/157 (59%)	44 (20–106)	46 (13–100)	81 (98)	80 (111)	2 (−31 to 35)	0.92
Hazardous alcohol use	67/152 (44%)	49 (21–98)	45 (16–111)	82 (107)	82 (103)	0 (−34 to 34)	0.99
Chronic kidney disease	23/156 (15%)	42 (17–73)	46 (18–109)	72 (140)	82 (96)	−11 (−57 to 36)	0.65
Chronic lung disease	29/155 (19%)	45 (11–103)	46 (19–100)	70 (83)	84 (108)	−14 (−56 to 28)	0.51
Immunosuppression	25/110 (23%)	50 (15–123)	49 (19–106)	94 (143)	82 (92)	13 (−35 to 60)	0.60
Malignancy	17/155 (12%)	41 (21–117)	46 (17–99)	74 (80)	82 (106)	−8 (−61 to 45)	0.76
No risk factors for melioidosis	11 (7%)	49 (13–61)	46 (18–104)	59 (75)	82 (104)	−23 (−87 to 40)	0.47
Lung involvement	120 (75%)	46 (19–102)	42 (14–96)	88 (113)	58 (50)	30 (−7 to 67)	0.12
SSTI	18 (11%)	62 (39–154)	45 (17–96)	124 (137)	75 (96)	49 (−1 to 99)	0.06 ^c^
Musculoskeletal involvement	31 (20%)	73 (38–144)	40 (17–88)	116 (115)	72 (98)	44 (4–84)	0.03 ^c^
CNS involvement	8 (5%)	44 (32–155)	46 (17–98)	83 (77)	80 (104)	3 (−70 to 77)	0.93
Genito-urinary involvement	34 (21%)	82 (19–106)	41 (18–97)	84 (88)	80 (106)	4 (−35 to 43)	0.84
Isolated bacteraemia	9 (6%)	38 (22–65)	46 (18–103)	47 (35)	82 (105)	−35 (−104 to 34)	0.32

No children had persistently positive blood cultures. IQR: interquartile range; SD: standard deviation; SSTI: skin and soft tissue infection; CNS: central nervous system. Median and mean are presented as the data had a non-parametric distribution. ^a^ Retrospective data collection from cases before October 2016 resulted in some missing data prior to this time and, accordingly, a difference in the denominator for some variables. ^b^ *p* value determined using linear regression. ^c^ Variables selected for the multivariable analysis.

**Table 5 tropicalmed-08-00481-t005:** Demographic and clinical characteristics of the patients with melioidosis who did and did not have a source control procedure.

	*n* = 477 (%) ^a^	Source Control *n* = 140	No Source Control *n* = 337	Odds Ratio (95% Confidence Interval)	*p* ^b^
Age ^c^	477	53 (39–61)	60 (43–67)	0.94 (0.89–0.99)	0.02 ^d^
Male gender	477	106 (76%)	227 (67%)	1.51 (0.96–2.37)	0.07 ^d^
First Nations Australians	477	75 (54%)	156 (46%)	1.34 (0.90–1.99)	0.15
Wet season presentation	477	101 (72%)	254 (75%)	0.85 (0.54–1.32)	0.46
Diabetes mellitus	237/460 (52%)	80/136 (59%)	157 (48%)	1.52 (1.01–2.28)	0.04 ^d^
Hazardous alcohol use	171/444 (39%)	57/134 (43%)	114/310 (37%)	1.27 (0.84–1.92)	0.25
Chronic kidney disease	59/458 (13%)	12/136 (9%)	47/322 (15%)	0.57 (0.29–1.10)	0.095
Chronic lung disease	95/447 (21%)	23/134 (17%)	72/313 (23%)	0.69 (0.41–1.17)	0.17
Immunosuppression	58/327 (18%)	16/96 (17%)	42/231 (18%)	0.90 (0.48–1.69)	0.74
Malignancy	47/446 (11%)	6/134 (4%)	41/312 (13%)	0.31 (0.12–0.74)	0.009 ^d^
No risk factors for melioidosis	73 (15%)	15 (11%)	58 (17%)	0.57 (0.32–1.06)	0.08 ^d^
Bacteraemic	332 (70%)	89 (64%)	243 (72%)	0.68 (0.44–1.03)	0.07
Lung involvement	336/466 (72%)	95/139 (68%)	241/327 (74%)	0.77 (0.50–1.19)	0.24
SSTI	70/459 (15%)	30/139 (22%)	40/320 (13%)	1.93 (1.14–3.25)	0.01 ^d^
Musculoskeletal involvement	58/459 (13%)	38/139 (27%)	20/320 (6%)	5.64 (3.14–10.1)	<0.0001 ^d^
CNS involvement	16/459 (3%)	5/139 (4%)	11/320 (3%)	1.05 (0.36–3.08)	0.93
Genito-urinary involvement	90/460 (20%)	51/139 (37%)	39/321 (12%)	4.19 (2.59–6.78)	<0.0001 ^d^
Septic shock	99/443 (22%)	35/134 (26%)	64/309 (21%)	1.35 (0.84–2.17)	0.21
ICU admission	118 (25%)	41 (29%)	77 (23%)	1.40 (0.90–2.18)	0.14
Died	55 (12%)	7 (5%)	48 (14%)	0.32 (0.14–0.72)	0.004

All numbers are presented as median (interquartile range) or the absolute number (%). SSTI: skin and soft tissue infection; CNS: central nervous system; ICU: intensive care unit. ^a^ Retrospective data collection from cases before October 2016 resulted in some missing data prior to this time and, accordingly, a difference in the denominator for some variables. ^b^ *p* value determined using univariable logistic regression. ^c^ Age (in years) divided by 5 for this analysis to facilitate interpretation. ^d^ Variables selected for the multivariable analysis.

## Data Availability

Data cannot be shared publicly because of the Queensland Public Health Act 2005. Data are available from the Far North Queensland Human Research Ethics Committee (contact via email FNQ_HREC@health.qld.gov.au) for researchers who meet the criteria for access to confidential data.

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
