# Peer review of "The Epidemiological, Clinical, and Microbiological Features of Patients with Burkholderia pseudomallei Bacteraemia—Implications for Clinical Management"

_tropicalmed, 2023, doi:10.3390/tropicalmed8110481_

Round 1

Reviewer 1 Report

Comments and Suggestions for Authors

Title: The epidemiological, clinical and microbiological features of patients with Burkholderia pseudomallei bacteraemia — implications for clinical management.

Manuscript ID: 2594738

I recommended that manuscript could be accepted with MINOR MODIFICATIONS. However, molecular studies could be impact in the study and conclusion related with the involved genotype (although authors mentioned this point as a limitation of the study).

ABSTRACT

-          Include method used to identify B. pseudomallei.

INTRODUCTION

-          Line 42, eliminate ¨and¨.

MATERIALS AND METHODS

-          Specify method used after blood culture to identify B. pseudomallei.

-          Mentioned that positive cases were management by doctors in the hospital.

GENERAL COMMENTS

-          Review typographic mistakes.

Reviewer 2 Report

Comments and Suggestions for Authors

The study investigates the prevalence of bacteremia among patients with infection by Burkholderia pseudomallei, confirmed by positive culture, admitted to Cairns Hospital and its satellite hospitals between January 1998 and June 2023. The presence of bacteremia was correlated with patient characteristics or laboratory data in order to seek clinically relevant information. Given the high rate of bacteremia in these patients, exploring characteristics associated with a worse outcome is of certain interest.

However, the manuscript needs to be extensively revised, both in terms of the organization of sections and the methodology used. Unfortunately, in its current state I cannot recommend proceeding with the revision process.

Please find below some suggestions:

The introduction is too scarce in justifying the study, while a full specter of reasons is given in the first part of the discussion. The latter, on the contrary, seems to delve too deeply into the clinical management issues of patients with melioidosis without having a reflection of findings of the study. Authors should rewrite the introduction to appropriately contextualize the presented study.

As mentioned above, the discussion is too lengthy (especially between lines 234-271) and does not allow for the capture of the new findings highlighted by the present study. Authors should move some argument to the introduction section and focus more on commenting the actual findings.

It seems that for bacteremia and source control, univariable logistic regressions were performed, but the relative tables presumably report the p value of the chi2 test. This information is redundant, so it would help clarity to report in both tables the ORs (with 95% CI and p-value) of the univariable logistic regressions, especially if the variable selection for the multivariable models has been made based on the latter.

In the methods section is stated that “Multivariate analysis was performed using backwards stepwise logistic regression with variables selected for inclusion in the model if their p value in univariate analysis was <0.10.” Have the p-values of the logistic regressions been taken into account, or those of the chi-squared/Fisher’s tests?

On the same line, while this is an accepted way for model building, the number of available patients among the different analyzed variables appears to be too ample to simply use this method. Authors should report the final sample used for the two logistic regressions for bacteriaemia and source control (i.e. specifying the number of patients for whom all data are available and included in the multivariate analysis, presumably ≤327). The authors should also report the univariate analyses conducted on these patients and should base the selection of variables to include in the multivariate models on these analyses.

The variable “no risk factors for melioidosis” should be explicitly defined in the methods section.

A list of the variables originally included in the two multivariable logistic regression models should be listed in the methods section.

Line 93: "Groups were analyzed using the chi-squared, Fisher’s exact, the Kruskal-Wallis tests, or logistic regression where appropriate." For clarity, it would be useful to specify in tables the used test for each of the variables, possibly in the form of footnotes to the p-value. Moreover, there do not appear to be opportunities to use the Kruskal-Wallis test. Also, no test for the analysis of continuous variables (i.e. age) is reported.

In the discussion section, the absence of an index of severity of clinical conditions, such as SAPSII or APACHE score, should be reported and discussed among limitations.

Reviewer 3 Report

Comments and Suggestions for Authors

Dear Authors and Editor,

Thank you for considering me to review this interesting article. The authors seek to further our clinical knowledge regarding the disease Melioidosis. The author’s assertion is correct that there needs to be a greater understand of this disease and clinical studies should play a role in this endeavour. They have consolidated a wealth of information concerning confirmed cases of melioidosis with an endemic region.

The manuscript is well written and clear.

I have some comments that I think should be addressed prior to publishing

·         The authors perform statistical analyses on four dependant variables (‘proportion with bacteraemia’, ‘TTP’, ‘duration of bacteraemia’ and ‘source control’). Multiple test are performed independently on the same data set. I feel that, unless these are variables are used in the same model (i.e. a regression with tens of explanatory variables – which will not work), a familywise error should be considered. The reader should be given an adjusted and unadjusted p-value.

·         Throughout the authors should state the test used. They state which are use “if appropriate” however the reader has no site on the diagnostic plots/statistics so it becomes impossible to know which tests have been used where. These details could be provide within the tables.

·         The authors need to be clear what the meaning is behind the results of the stepwise regression. This very powerful technique has iteratively found the best way to describe the data with the least number of variates. This means that those variables that did not make it into the multiple regression model, but independently seemed to associate with the dependant variable, are likely correlates to those that did. The author should take each of the four variables in the multiple regression model and consider how they relate other variables that were not included in the model. In this way, the reader can understand the (often-overlapping) relationship between the variables. One option might be to use cluster analysis and dendrogram to represent the relationship between explanatory variables. Another option might be a correlation matrix. These analyses might not need to be included within the body of the manuscript (i.e. be supplemental) but could be referred to.

·         Why was a multivariate approach taken for the dependant variable ‘proportion with bacteraemia’ but not ‘TTP’, or ‘duration of bacteraemia’ or ‘source control’? I think a stepwise approach should be used throughout. Various methods do exist that would perform stepwise analysis on scaled data – such as multiple regression.

Round 2

Reviewer 2 Report

Comments and Suggestions for Authors

The authors have fully addressed the comments, and I am pleased to recommend the paper for publication.

Author Response

We thank the reviewer for the time that he/she has taken to review the original and revised version of our manuscript. We are heartened that we have addressed all his/her concerns and that he/she recommends the paper for publication.

Reviewer 3 Report

Comments and Suggestions for Authors

Dear authors and editor,

First thank you to the authors for taking my comments seriously. I continue to think that this work is valuable and think that this manuscript is a well written and interesting investigation. I am also glad that the authors have brought in an additional statistician. Other than a single point that I would like the authors to consider further, I am happy for this work to be published.

I am still a little troubled by authors statement “To retain power, no adjustments were made to significance levels for multiple comparisons”. Obviously this statement is true but it contains no reason to accept a raised false positive rate associated with multiple tests on the same data. The only possible reason accept a right risk of false positives is that you don’t want to miss something interesting. I accept that this study is an exercise data exploration so perhaps I am being too rigid in the application of probability. Moreover the authors use pan-explanatory variable stepwise methods to consider each dependant variable and these are not susceptible to familywise error. Perhaps I can ask the authors to expand on this statement? Statement such as (to paraphrase) “Multiple tests using the single explanatory variables and dependant variable were not adjusted for familywise error. We therefore accept an increased risk of a false positive. However it is important for these exploratory analyses to consider all possible factors that relate to disease and not be penalised for the breadth of factors considered. Findings were strengthened in that some factors were later verified by multivariate analysis.”

Best wishes,

Reviewer 3

Author Response

We thank the reviewer for the time that he/she has taken to review the original and revised version of our manuscript. We are heartened that we have addressed  all but one of his/her concerns.

We agree that we could have emphasised the risk of family wise error more in our manuscript.  In the revised manuscript we have exanded the limitations section to address this point more thoroughly in a manner suggested by the reviewer.

"In order to retain power, we didn’t adjust significance levels for multiple comparisons which increases the risk of family wise error. However, this approach allowed the examination of multiple potential associations in this exploratory analysis, which were then evaluated further in multivariable analysis. The provision of the raw data allows the reader – and other clinicians – to assess the biological plausibility of our findings" (lines 299-303)